# A mixed method evaluation of a theory based intervention to reduce sedentary behaviour in contact centres- the stand up for health stepped wedge feasibility study

**Divya Sivaramakrishnan** [1] *, **Graham Baker**[2], **Richard A. Parker**[3], **Jillian Manner**[1], **Scott Lloyd**[4,5,6], **Ruth Jepson**[1]

1 Scottish Collaboration for Public Health Research and Policy, University of Edinburgh, Edinburgh, United Kingdom, 2 Physical Activity for Health Research Centre, University of Edinburgh, Edinburgh, United Kingdom, 3 Edinburgh Clinical Trials Unit, Usher Institute, University of Edinburgh, Edinburgh, United Kingdom, 4 Public Health South Tees, Middlesbrough Council and Redcar & Cleveland Borough Council, Middlesbrough, United Kingdom, 5 Fuse–UKCRC Centre for Translational Research in Public Health, Population Health Sciences Institute, William Leech Building, Newcastle University, Newcastle upon Tyne, United Kingdom, 6 Teesside University, Middlesbrough, United Kingdom

* divya.sivaramakrishnan@ed.ac.uk

**Data Availability Statement:** The Stand Up for health Questionnaire data (anonymised) has been published on University of Edinburgh server:

## Abstract

### Introduction

Contact centres have higher levels of sedentary behaviour than other office-based workplaces. Stand Up for Health (SUH) is a theory-based intervention developed using the 6SQuID framework to reduce sedentary behaviour in contact centre workers. The aim of this study was to test acceptability and feasibility of implementing SUH in UK contact centres.

### Methods

The study was conducted in 2020–2022 (pre COVID and during lockdown) and used a stepped-wedge cluster randomised trial design including a process evaluation. The intervention included working with contact centre managers to develop and implement a customised action plan aligning with SUH's theory of change. Workplace sedentary time, measured using activPAL™ devices, was the primary outcome. Secondary outcomes included productivity, mental wellbeing, musculoskeletal health and physical activity. Empirical estimates of between-centre standard deviation and within-centre standard deviation of outcomes from pre-lockdown data were calculated to inform sample size calculations for future trials. The process evaluation adopted the RE-AIM framework to understand acceptability and feasibility of implementing the intervention. Interviews and focus groups were conducted with contact centre employees and managers, and activity preferences were collected using a questionnaire.

Sivaramakrishnan, D. (Creator) & Jepson, R. (Creator), Edinburgh DataShare, 8 Dec 2021 DOI: 10.7488/ds/3250.

**Funding:** This project is funded by the National Institute for Health Research (NIHR) [PHR project grant: 17/149/19]. The views expressed are those of the authors(s) and not necessarily those of the NIHR or the Department of Health and Social Care. https://www.nihr.ac.uk/ RP was partially supported in this study by NHS Lothian via the Edinburgh Clinical Trials Unit. For the purpose of open access, the author has applied a Creative Commons Attribution (CC BY) licence to any Author Accepted Manuscript version arising from this submission. The funders had no role in study design, data collection and analysis, decision to publish, or preparation of the manuscript.

**Competing interests:** The authors have declared that no competing interests exist.

## Results

A total of 11 contact centres participated: 155 employees from 6 centres in the pre-lockdown data collection, and 54 employees from 5 centres post-lockdown. Interviews and focus groups were conducted with 33 employees and managers, and 96 participants completed an intervention activity preference questionnaire. Overall, the intervention was perceived as acceptable and feasible to deliver. Most centres implemented several intervention activities aligned with SUH's theory of change and over 50% of staff participated in at least one activity (pre-lockdown period). Perceived benefits including reduced sedentary behaviour, increased physical activity, and improved staff morale and mood were reported by contact centre employees and managers.

## Conclusions

SUH demonstrates potential as an appealing and acceptable intervention, impacting several wellbeing outcomes.

## Trial registration

The trial has been registered on the ISRCTNdatabase: http://www.isrctn.com/ISRCTN11580369.

## Introduction

Sedentary behaviour in adults is linked with an increased risk of cardiovascular and all-cause mortality, type 2 diabetes, cardiovascular disease, musculoskeletal health issues, and poor mental health [1–5]. Physical activity can modify the associations between health risks and sedentary behaviour, with high levels of moderate to vigorous physical activity offsetting the mortality risks associated with high levels of sedentary behaviour [1, 6, 7]. However, recent evidence on the prevalence of physical activity suggests that it would be difficult to shift the population to meet the high levels of physical activity (> 300 min/week) required for this [1], and therefore distinct strategies to reduce sedentary behaviour are still required.

Contact centres are workplaces where employees handle a high volume of incoming and/or outgoing calls for various purposes such as sales and customer service [8]. They are currently considered one of the most sedentary working environments and are associated with higher levels of sedentary behaviour as compared to other office-based environments [9–11]. In a sample of Australian employees, it was found that contact centre staff were sedentary for a significantly higher proportion of their working time (83.4%) compared to other office workers (75.8%) [9]. High levels of sedentary behaviour are associated with musculoskeletal pain [4], and studies have reported that 60% to 65% of contact centre employees experience such issues [12, 13]. Due to technological, environmental, and cultural factors, contact centre staff work within constrained conditions—restricted to their desks making it difficult for them to move regularly. Staff report low workplace autonomy, performance monitoring, and poor job security [14]. The fast-paced nature of contact centre work and organisational pressures to maintain high levels of productivity and profitability mean that investment into health and physical activity programmes are often de-prioritised [15]. Therefore, contact centres present a unique setting to develop sedentary behaviour interventions which benefit employee health but recognise the needs of the employer to remain productive.

To date, there are no published systematic reviews assessing the effectiveness of sedentary behaviour interventions in contact centres and very few interventions have been developed and evaluated in this setting [16–19]. Two pilot studies conducted in the US [18] and Australia [19], evaluated the effectiveness of using standing desks in reducing sedentary time. Pickens et al. reported significant effects at three and six month follow up [18], and Chau et al. reported that intervention participants increased standing time at one and four weeks [19]. Two studies were conducted in the UK [16, 17], including a non-randomised, pre-post study evaluating the feasibility of a multi-component intervention (including height-adjustable workstations, emails, education, training sessions, support from team leaders and a workplace champion) targeting prolonged sitting in contact centres [16]. The study concluded that the intervention was largely perceived positively, and identified education sessions, height-adjustable workstations and emails as intervention components that were regarded as the most effective. Limitations of these studies include: the use of a single component intervention (standing desks) [18, 19]; small sample sizes; and short follow up periods [16]. One randomised controlled trial with a sample size of 59 contact centre agents compared two interventions including multi-component strategies (with and without height adjustable workstations) delivered over 10 months [17]. While the study found that the intervention with height adjustable workstations reduced worktime sitting compared to the other group, a limitation of the study was the lack of a true control group. A systematic review of white-collar workers (who work in offices and are not involved in manual labour) found that multi-component interventions (i.e., an instalment of sit-stand workstations in combination with behavioural interventions such as goal setting) are most effective in reducing sitting time [20]. A recent rapid review identified 22 studies of randomised controlled trials that aimed to reduce sedentary behaviour in office settings, with five studies conducted in the UK [21]. However, there is a lack of multi-component interventions focussing or the reduction of sedentary behaviour among contact centre employees with rigorous development processes and evaluation designs.

Stand Up for Health (SUH) is a complex intervention that aims to reduce sedentary behaviour in contact centres and takes into account the context and needs of employees and employers. The SUH programme was developed through a rigorous process using the 6SQuID intervention development framework, as discussed in an earlier paper [22]. It is an adaptive intervention based on the Social Cognitive Theory [23] and the Social Ecological Model (SEM) [24]. The theories of change at the heart of the programme focus on organisational, environmental, social/cultural, and individual levels, as well as creating a sense of ownership among contact centre employees and employers. It also recognises the need to increase knowledge and awareness of the risks of sedentary behaviour to impact both health and work-related outcomes such as absenteeism and productivity. The programme prioritises fidelity to these theories of change rather than the specific activities. For example, in one centre, environmental change may take the form of standing desks, whilst in another it may be change in usage of space. The 6SquID framework consists of six steps including 1. Defining the problem, 2. Identifying modifiable and non-modifiable causal factors, 3. Defining the theory of change, 4. Defining the theory of action, 5. Testing and refining the intervention and 6. Collecting evidence of effectiveness to justify evaluation and implementation [25, 26]. Steps 1 to 4 and part of step 5 have been carried out as part of this programme of work and reported elsewhere [22].

Step 5 of the 6SquID framework involves testing and adaptation of the intervention [25, 26]. Testing could take the form of feasibility studies, seeking to understand whether the study can be carried out. Eldridge et al. (2016) describe a feasibility study as one that seeks to understand if something (project/development/future study) can be done, whether it should be taken forward and how to proceed [27]. This phase could test the acceptability of the intervention, recruitment methods, the theory of action, fidelity to the theory of change, and if there

are any unintended consequences [25, 26]. Feasibility testing relating to both the intervention and evaluation design is also recommended by the MRC framework for developing and evaluating complex interventions [28]. Eldridge et al. (2016) conceptualise pilot studies as a subset of a feasibility study, where a future study, or part of a future study is conducted on a smaller scale [27]. Step 5 of 6SQuID, testing and adapting the intervention, was started during the development and early piloting stage [22]. To complete this crucial step and add to the evidence to justify evaluation and implementation (Step 6), this SUH feasibility study with a nested pilot component was conducted. This paper summarises the findings of this multi-centre, feasibility study incorporating a stepped wedge trial design and process evaluation. A full report of the evaluation is also available [29].

The aim of this study was to test the acceptability and feasibility of implementing the SUH intervention in UK contact centres [30]. Aspects relating to the acceptability and feasibility of the research design and data collection procedures will be reported separately. This paper will report on the following study objectives:

1. Provide estimates of the variability of primary outcome (sedentary behaviour in the workplace) and secondary outcomes to inform sample size calculations for future studies

2. Explore experiences and acceptability of the SUH intervention activities and implementation processes

3. Understand whether SUH activities worked as intended and investigate any unintended consequences of the intervention

4. Identify differences in delivery of the intervention, between different contact centres and the reasons for these.

## Methods

The Consort checklist for pilot and feasibility studies (S1 Checklist) has been adhered to while reporting on this study [31]. Ethical approval for the project was received from the School of Health in Social Sciences Ethics Committee (University of Edinburgh, Ref: STAFF142).

### Study design

The SUH study comprised of a preliminary outcome evaluation, undertaken to address objective 1, and a process evaluation to address objectives 2–4. The study used a stepped wedge cluster randomised feasibility trial design [30]. Eleven contact centres were recruited and randomised to one of five unique sequences (Fig 1), with each sequence corresponding to specific intervention start dates and data collection time points. Data collection for the outcome evaluation was scheduled at the end of the control period (baseline), post-intervention period, and also three months after the end of the post-intervention period (objective 1). Focus groups and interviews for the process evaluation were scheduled to be conducted three months after the intervention period (objectives 2–4). Detailed information on the study design and rationale are reported elsewhere [30], and the original study protocol is attached as a S1 Protocol. The trial has been registered on the ISRCTNdatabase: http://www.isrctn.com/ISRCTN11580369.

**Impact of COVID-19.** The COVID-19 pandemic impacted the lives of billions across the globe, and also disrupted implementation of the SUH study. The lockdown measures introduced by the UK government in March 2020 affected contact centre operations, and several centres adopted work from home or hybrid formats. The measures also meant that the SUH

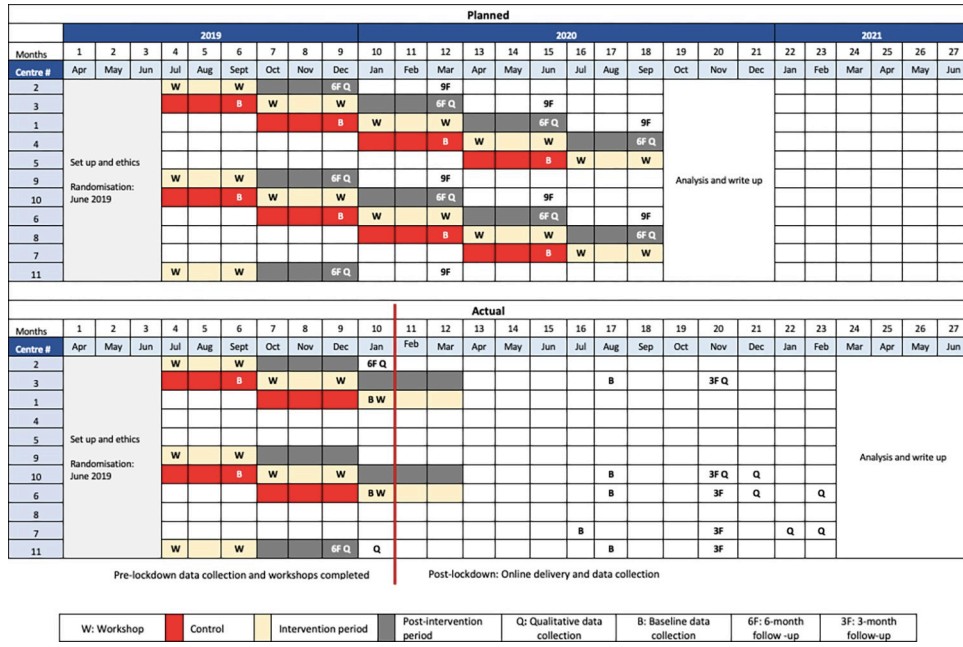

**Fig 1. Stand up for health cluster trial design- planned and actual.** Adapted from SUH NIHR report [29].

team could no longer make in-person visits. The study followed the stepped wedge schedule from July 2019 until January 2020, after which the design schedule, intervention delivery and data collection had to be altered and adapted (Fig 1). In this context, we will report the study methods and results for the pre and post lockdown periods separately. The post-lockdown study schedule was altered with outcome data collection being conducted before and after the 3-month intervention, and some focus groups and interviews were scheduled at the post-intervention timepoint (Fig 1).

## Participants and randomisation

Eleven contact centres from across England and Scotland were recruited to participate in the study. Randomisation of contact centres to sequences was conducted in May 2019, using computer-generated block randomisation, stratified by centre size ($\leq$ 500 employees versus > 500 employees). Randomisation was conducted by the project statistician, who was fully blinded to the names of the contact centres, and who generated a list of centre numbers showing the sequences that each centre should be allocated to. Centres were contacted by the SUH team approximately three months before the start date to plan for programme delivery or baseline data collection. Prior to this, centres were unaware of when they would start the intervention.

We aimed to recruit 27 participants from each centre for the outcome evaluation, and conduct focus groups with 6–8 employees, and interviews with those in relevant managerial positions (main point of contact for the SUH team) for the process evaluation. Sample size calculations and rationale are reported elsewhere [30]. All participants received the study information sheet and provided written or online consent (through the qualtrics[xm] platform) prior to participation in any form of data collection.

**Participant recruitment for preliminary outcome evaluation.** For the pre-lockdown intervention, the project team coordinated with centre managers to recruit participants. Recruitment strategies included posters, recruitment videos, and in-person recruitment visits

to centres by the SUH team. Inclusion criteria stated that participants needed to be staff of working age (16 years or older) who were provisionally scheduled to work for any amount of time during the seven days of collection of objectively measured sedentary behaviour data. For the post-lockdown programme, the project team coordinated with the centre managers to schedule a 20-minute consultation with staff members. In some centres, a doodle poll was used to schedule the consultation sessions directly with participants. All those who participated in the consultations were requested to complete the SUH questionnaire as a part of the outcome evaluation. Participants were given a £5 Love2Shop voucher at each data collection time-point after the completion of outcome data collection procedures.

**Participant recruitment for process evaluation.** Most focus groups and interviews with employees (in-person and online) were arranged by the centre manager. For some online focus groups and interviews, the SUH team liaised directly with the employees to arrange the session. Interviews with managers from six centres were conducted over the phone and through Microsoft (MS) Teams. Participants were given a £5 Love2Shop voucher after participation in interviews or focus group discussions.

## Intervention

**Pre-lockdown intervention.** SUH is a multi-component, adaptive programme that targets the organisational, environmental, social and individual levels, considering the system, cultural and environmental constraints of each contact centre. It was designed to be scalable and transferable to other contact centres and workplace settings. The programme was operationalised to co-produce the specific intervention activities and included three main elements. Firstly, a workshop where staff tried out various equipment and activities and participated in a prioritisation exercise to express their preferences for individual, social, and environmental activities. The SUH team loaned several pieces of equipment (exercise bike, mini table tennis etc) to the centres, and Sit-Stand.Com® provided desk risers to centres at no cost. A second workshop was held after three months, where the SUH team spoke to the staff about activity implementation, likes and dislikes, and suggestions to ensure staff involvement and ownership. Secondly, setting up of a SUH committee of staff members from teams across the participating centre. The committee was an essential element of the programme and was responsible for seeking and generating ideas for activities from staff and aiding implementation. Thirdly, a centre specific action plan was developed by the SUH team and centre managers. A "SMART" (Specific, Measurable, Achievable, Relevant, and Time-Bound) approach was adopted to enhance success of implementation, and the adoption of at least one activity from each level of the theory of change was encouraged. The intervention duration was approximately three months, during which time the centres developed their preferred activities for each theory of change, prepared an action plan for sustained engagement and tested out some of the activities. A detailed intervention guide has been provided as a S1 File.

**Post-lockdown intervention.** During the lockdown period, the SUH team was unable to conduct in-person workshops, and instead conducted one-on-one virtual consultations with up to 30 staff per centre. Each consultation session lasted for approximately 20 minutes, where barriers/facilitators to sedentary behaviour and physical activity at the various levels were discussed. The team then provided each participant with an individually tailored plan with recommendations and resources (example plan attached: S2 File). Based on the consultations, the team also generated a general plan for the contact centre which could be shared with all staff members. The SUH team sent some equipment (for example: balance board, balance ball chair, mini table tennis, twisting discs) to one centre, for the benefit of staff who were working on site. In addition, the team organised a step count challenge, conducted over six weeks, to

**Table 1. Stand up for health outcomes for the pre and post lockdown periods.**

| Outcome | Instrument/tool | Measured at pre-lockdown | Measured at post-lockdown |
|---|---|---|---|
| **Primary outcome** | | | |
| Objectively measured sedentary time in the workplace | activPAL™ device | ✓ | |
| **Secondary outcomes** | | | |
| Subjectively measured sedentary time in the workplace | The Occupational Sitting and Physical Activity Questionnaire (OSPAQ) | ✓ | ✓ |
| Objectively measured prolonged sitting time in the workplace (bouts of > = 30 minutes) | activPAL™ device | ✓ | |
| Objectively measured total sedentary time (i.e. including time outside the workplace such as at home and leisure time) | activPAL™ device | ✓ | |
| Objectively measured workplace and total standing time | activPAL™ device | ✓ | |
| Objectively measured workplace and total physical activity (based on stepping) | activPAL™ device | ✓ | |
| Objectively measured workplace and total sit-to-stand transitions | activPAL™ device | ✓ | |
| Productivity | Utrecht Work Engagement Scale (UWES) | ✓ | ✓ |
| Mental wellbeing | Warwick-Edinburgh Mental Well-being scale (WEMWBS) | ✓ | ✓ |
| Musculoskeletal health | MSK-HQ | ✓ | ✓ |
| Physical activity (meeting physical activity recommendations) | Scottish Physical Activity Screening Question (Scot-PASQ) | ✓ | ✓ |
| Participation in activities and preferences | Activities Questionnaire | ✓ | ✓ |
| Staff turnover | Number of people leaving and number of new joiners over the study period | ✓ | ✓ |

Note: ✓ indicates that the outcome was measured during the specified period

encourage staff to sit less, move more and generate social interaction. Staff formed teams of five and made a virtual trip from Land's End to John O'Groats (an iconic long-distance cycle ride that spans the length of mainland UK). They submitted the weekly steps for their team on the SUH website.

## Outcomes

The study's primary and secondary outcomes, and whether they were measured during the pre and post lockdown periods are presented in Table 1. Additional outcomes relating to absentee-ism, call times and sick leave are presented in the full report of this evaluation [29].

## Process evaluation RE-AIM framework

The RE-AIM framework was used to guide the process evaluation [32, 33]. A summary of the framework and measurement for process evaluation, for the pre and post lockdown periods is presented in Table 2. The Reach dimension of the framework explored whether the interven-tion was available to everyone within each contact centre, and also captured the appeal and acceptance of the programme. Activity preferences among staff were also presented under the Reach section to further understand programme appeal. The Effectiveness element examined perceived benefits and consequences of the programme among staff and managers. Since this was a feasibility study, Adoption (the percentage of contact centres that participated in the SUH programme) has limited relevance, and the proportion of centres that participated out of those that were targeted was assessed. The Implementation element examined the activities implemented by the contact centres. The SUH intervention is an adaptive programme that

**Table 2. Process evaluation framework.**

| RE-AIM dimension and definition | Pre-lockdown elements | Pre-lockdown measurement | Post-lockdown elements | Post-lockdown measurement |
|---|---|---|---|---|
| **Reach:** Availability, acceptance, and appeal of the SUH programme | • Programme significance and appeal • Programme participation • Barriers & enablers | All aspects were explored qualitatively. Programme participation includes data on activity preferences among staff captured using the SUH questionnaire | • Programme significance and appeal • Programme participation • Barriers & enablers | All aspects were explored qualitatively. Programme participation includes data on activity preferences among staff captured using the SUH questionnaire. |
| **Effectiveness:** Perceived benefits of the SUH programme | • Perceived benefits | Qualitative data on perceived benefits from interviews/focus groups with managers and staff (Note: quantitative results from outcome evaluation presented in a separate section) | • Perceived benefits | Qualitative data on perceived benefits from interviews/focus groups with managers and staff (Note: quantitative results from outcome evaluation presented in a separate section) |
| **Adoption:** Proportion of centres that participated out of those that were targeted for recruitment | | Number of participating centres /Number of targeted centres | | |
| **Implementation:** Programme theory elements and unintended consequences | • Organisational level • Environmental level • Social level • Individual level • Ownership • SB and PA awareness • Unintended consequences | Explored through qualitative interviews/ focus groups with managers and staff | • Organisational level • Environmental level • Social level • Individual levels • Ownership • SB and PA awareness • Unintended consequences | Explored through qualitative interviews/ focus groups with managers and staff |
| **Maintenance:** Contact centre plans to continue with the programme | | | • Future of SUH | Explored through qualitative interviews/ focus groups with managers and staff |

Adapted from SUH NIHR report [29]

does not prescribe specific activities, to allow for flexibility, scalability and transferability. Therefore, the programme implementation could vary between centres, with each choosing different activities within the organisational, environmental, social and individual levels. Hence, rather than assessing fidelity to specific activities or consistency of delivery across sites, the process evaluation aimed to verify the programme's theory of change. Accordingly, to assess implementation, the programme theory elements (organisational, environmental, social and individual factors, ownership over the programme, and awareness of physical activity and sedentary behaviour), as well as unintended consequences were explored during the focus groups and interviews. The Maintenance aspect covered contact centre plans to continue with the SUH programme.

## Data collection

**Outcome evaluation.** Table 3 shows details regarding data collection times and intervention start times with corresponding centre numbers.

During the pre-lockdown period, in-person visits were made by the SUH team to contact centres where they collected activPAL™ and questionnaire data. Participants received a pack consisting of an activPAL™, a logbook to capture work and sleep times (S3 File), two alcohol wipes, and two Tegaderm (self-adhesive skin dressing) strips. The pack contained an additional alcohol wipe and Tegaderm for participants to reattach the activPAL™ in case it became

**Table 3. Data collection time points.**

| Timepoint for data collection | Pre/Post lockdown period | Centre numbers | Started intervention | Baseline/Outcome data collected | Notes |
|---|---|---|---|---|---|
| **A: Dec 2019/Jan 2020** | Pre | 2, 11 | July 2019 | Dec 2019/ Jan 2020 | Follow-up outcome data collected only |
| **B: Sep 2019** | Pre | 3, 10 | Oct 2019 | Sep 2019 | Baseline data collected only |
| **C: Jan 2020** | Pre | 1, 6 | Jan 2020 | Jan 2020 | Baseline data collected only |
| **D: Jul/Aug 2020** | Post | 3, 6, 7, 10, 11 | Jul/Aug 2020 | Jul/Aug 2020 | Baseline data collected |
| **E: Nov 2020** | Post | 3, 6, 7, 10, 11 | Jul/Aug 2020 | Nov 2020 | 3 months follow-up |

detached. The SUH team briefed participants on how to attach the activPAL™, and attached the activPAL™ for participants who requested help. Participants were requested to wear the activPAL™ and complete the logbook over a 7-day period. Online questionnaires were created using the qualtrics^xm platform, and participants were requested to complete them on tablets. Paper questionnaires were used if the tablet malfunctioned or if there were internet issues. Post-lockdown data collection included only online questionnaires circulated to participants through email by the SUH team or centre managers. Two researchers had access to identifying information during data collection. The data was stored in encrypted folders on university servers and only RJ, DS and JM had access to non-anonymised data. Data was anonymised and shared with the project statistician. At the end of the project, anonymised data was uploaded to the University of Edinburgh repository [34].

## Process evaluation

Topic guides for the focus group discussions and interviews were developed based on the process evaluation framework (Table 2), and covered topics relating to the acceptability and feasibility of the SUH intervention, and programme theory elements. The topic guides were amended for the post-lockdown period to include questions pertaining to post-lockdown activities (consultations, activity plan, and step count challenge). Focus groups were conducted in person before lockdown, and online during the post-lockdown period. Manager interviews were conducted over the phone or online. All focus group discussions had a moderator and co-moderator. The face-to-face focus groups and telephone interviews were recorded using an audio recorder. The online sessions were recorded using the record function on MS Teams.

## Statistical analysis

Descriptive statistics were used to summarise participant demographic details (age, gender, job title, employment type, length of time working at the contact centre, any health problems reported) for all participants and also stratified by contact centre. Categorical variables were summarised using number (%) and continuous variables using mean, median, standard deviation, minimum, maximum, and interquartile range. In addition, descriptive statistics of continuous participant outcomes were also produced and split by data collection time point. For the binary outcome of "meeting physical activity guidelines" according to the Scot-PASQ, frequencies and percentages were calculated.

Box and dot plots were used to present graphical representations of the data using ggplot 2 in R software [35]. Using the pre-lockdown data, empirical estimates of between-centre standard deviation and within-centre standard deviation of outcomes were calculated to inform sample size calculations for future cluster randomised trials. For the post-lockdown data, in order to distinguish between centre-level and participant-level effects, linear mixed models were fitted to each outcome variable, adjusting for participant and centre as nested random

effects. Model based estimates of between-centre, between-participant, and residual standard deviations were then extracted from the model results.

The major changes to the study design resulting from the impact of the COVID-19 pandemic meant that we could not fully adhere to the statistical analysis plan [30]. In particular, although we pre-specified sophisticated linear mixed models to produce preliminary estimates of the intervention effect [30], these are not reported in this publication because of concerns about their reliability due to the limited number of clusters and sequences available for analysis and potential confounding, leading to difficulties in interpretation. Full details about the linear mixed effects analysis can be found in the full NIHR report [29].

Three of the centres recruited during the pre-lockdown period provided information on rates of staff turnover between the pre- and post-lockdown periods (over a 10–12 month period). Proportions of participants leaving the company or moving jobs within the same company were computed with exact 95% confidence intervals. Similarly, for the post-lockdown period, the number of participants involved in the post-lockdown data collection who left the company or moved jobs over the 3 months follow-up period were calculated.

Most statistical analyses were performed using SPSS software version 24 [36], although R software version 4.0.4 [37] was used for graphics and processing the raw ActivPAL data via the "activpalProcessing" package [38].

**Process evaluation analysis.** Focus group and interview data were transcribed by a transcription agency. Transcripts were analysed using a codebook thematic analysis approach, where themes were identified by the researchers based on a predetermined coding framework [39, 40]. The coding framework was developed by DS, JM and GB based on the process evaluation framework (Table 2). Five transcripts were coded by both DS and JM (including a mix of pre and post lockdown, and staff and managers) based on the predefined codes. The other transcripts were coded by one researcher (DS/JM). Transcripts were coded deductively based on the coding framework to capture themes within the broad framework. Differences between centres within each theme were examined during analysis. A computer software package (Nvivo 11 for Windows) was used to code the transcripts and manage the thematic structure. GB and RJ acted as critical friends, who discussed the themes and subthemes with DS and JM, clarified and offered interpretation and provided insights and suggestions to refine the themes [41]. Activity preferences reported in the SUH questionnaire were summarised and reported as percentages.

## Results

### Centre characteristics

Eleven centres were recruited from across the United Kingdom. The centre size varied from 33–2000 staff members, with an average of 559 (SD 660) staff members.

### Preliminary outcome evaluation results

For the pre-lockdown data collection phase, there were 155 participants from six centres (Fig 2), with an average of approximately 26 participants per centre (range 21–33), of which approximately 16 per centre (range 3–25) provided valid primary outcome data, and 25 per centre on average (range 20–33) provided valid secondary outcome data (i.e. data from the OSPAQ questionnaire). The reasons for participants not providing valid primary outcome data were missing logbook information (n = 14), device fault (n = 4), device removed by participant (n = 2), device lost by participant (n = 1) and participant not at work (n = 1). For the post-lockdown period, 54 participants from five centres provided data for the preliminary

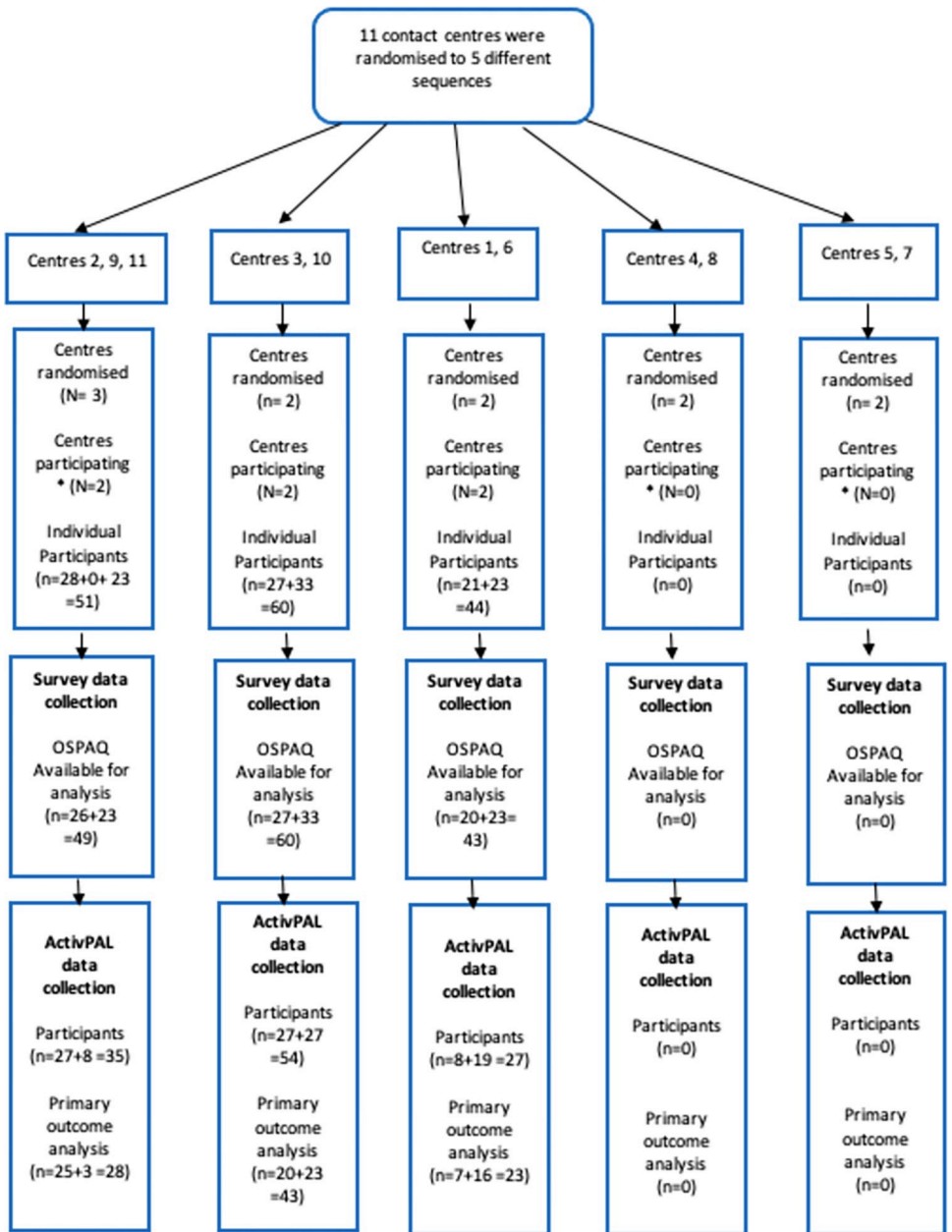

**Fig 2. CONSORT flow diagram for pre-lockdown data collection.** Adapted from SUH NIHR report [29]. *Centres 8 and 9 ceased participation. Centres 4 and 5 dropped out due to COVID-19. Centre 7 participated in the post-lockdown intervention.

outcome evaluation. The CONSORT flow diagram for the post-lockdown period is attached as a S1 Fig.

## Participant characteristics at baseline

A total of 155 participants took part in pre-lockdown and 54 participants took part in post-lockdown data collection. Most participants were aged 25–54 and worked full time. Pre-lockdown, 66% of participants were female and 34% were male. Post-lockdown, 54.9% were female

and 45.1% were male. Pre-lockdown, 51% were call handlers/customer service advisors, 24.5% were supervisors/managers/team leaders and 24.5% had other roles. During the post-lockdown period, most participants (86.3%) were call handlers/customer service advisors, 9.8% were supervisors/managers/team leaders and 3.9% had other roles. Detailed participant demographic details for the pre-lockdown and post-lockdown data collection periods are summarised using descriptive statistics in Tables 4 and 5 respectively.

**Descriptive analysis of outcomes.** Table 6 shows descriptive statistics of outcome measures for both the pre-lockdown and post-lockdown data collection periods. Please refer to Table 3 for details on what the time points refer to.

In the pre-lockdown phase, 97 out of 151 participants (64%) met the UK physical activity guidelines (range 57% to 79% in each centre). In the post-lockdown phase, 76% of participants (19/25) met UK physical activity guidelines at baseline, of which four no longer met physical activity guidelines at three months follow-up. Of the six participants not meeting physical activity guidelines at baseline, three of those proceed to meet physical activity guidelines at 3-months follow-up.

Fig 3 shows a box and dot plot of the primary outcome (sedentary time in the workplace) stratified by centre in the pre-lockdown phase, Fig 4 shows the same outcome stratified by data collection group (A, B, C) and Fig 5 shows self-reported sedentary time stratified by data collection group.

**Sample size calculations for future study.** The overall within-centre standard deviation (SD) of the primary outcome at baseline was 96 minutes (1.6 hours), and between centre SD was 44 minutes (0.7 hours). In addition, the mean value of the primary outcome at baseline was 359.6 minutes, and we recruited a mean of 15.6 participants providing valid primary outcome data per cluster. Therefore, using the equation (6) shown in Hayes and Bennett [42], we calculate that we need 25 clusters in a future cluster randomised trial to detect a clinically relevant difference in sedentary time of 45 minutes with 90% power and two-sided 5% significance level. We suggest that a future cluster trial, with clusters recruited in pairs, would have a low cluster-level drop-out rate of 10%, because in this study design clusters would not have to wait long to begin the intervention. We therefore, suggest that a future trial should aim to recruit 28 clusters in total, taking centre-level drop-out rates into account. We also recommend that a future trial aims to recruit approximately 40 participants per centre on average to use the ActivPALs in order to have at least 15–16 participants per centre recording valid primary outcome data.

In contrast, the subjective outcome measure of sedentary behaviour (the OSPAQ, measuring number of hours sitting) showed substantial variability in our study. The overall within-centre SD at baseline was 7.2 hours and between-centre SD was 2.3 for the pre-lockdown data. In the post-lockdown data, the between-centre SD was 5.2 hours with a between-participant SD of 6.8 hours.

**Staff turnover.** There were 13 out of 106 participants (12%, 95% CI 7 to 20%) recorded as having left the company in the 10–12 months period between the pre-lockdown and post-lockdown data collection periods. There were an additional 10 participants moving departments within the same company, and so the overall proportion of participants moving jobs and/or leaving the company was 23/106 (22%, 95% CI 14 to 31%).

During the post-lockdown phase, two participants from one centre (Centre 7, n = 20) left the company between baseline and follow-up three months later. In addition, two participants from another centre (Centre 10, n = 18) left the company and one moved departments within the same company. This equates to a staff turnover rate of approximately 10–20% in each of these two centres. Overall, five out of 54 participants left or moved after the baseline post-lockdown data collection (9%, 95% CI 3 to 20%).

**Table 4. Baseline demographics for participants involved in pre-lockdown data collection, stratified by contact centre.**

| | | Centre | | | | | | Total |
|---|---|---|---|---|---|---|---|---|
| | | 1 | 2 | 3 | 6 | 10 | 11 | |
| **Total number of participants** | | 21 | 28 | 27 | 23 | 33 | 23 | 155 |
| Age (years) [n = 153] | 18–24 | 8 | 0 | 2 | 0 | 1 | 3 | 14 |
| | | 38.1% | 0.0% | 7.4% | 0.0% | 3.1% | 13.0% | 9.2% |
| | 25–34 | 6 | 5 | 16 | 7 | 9 | 8 | 51 |
| | | 28.6% | 18.5% | 59.3% | 30.4% | 28.1% | 34.8% | 33.3% |
| | 35–44 | 2 | 6 | 4 | 8 | 13 | 9 | 42 |
| | | 9.5% | 22.2% | 14.8% | 34.8% | 40.6% | 39.1% | 27.5% |
| | 45–54 | 4 | 9 | 5 | 4 | 4 | 1 | 27 |
| | | 19.0% | 33.3% | 18.5% | 17.4% | 12.5% | 4.3% | 17.6% |
| | 55–64 | 1 | 7 | 0 | 3 | 4 | 1 | 16 |
| | | 4.8% | 25.9% | 0.0% | 13.0% | 12.5% | 4.3% | 10.5% |
| | > 65 | 0 | 0 | 0 | 1 | 1 | 1 | 3 |
| | | 0.0% | 0.0% | 0.0% | 4.3% | 3.1% | 4.3% | 2.0% |
| Gender [n = 153] | Male | 15 | 10 | 11 | 3 | 10 | 3 | 52 |
| | | 71.4% | 37.0% | 40.7% | 13.0% | 31.3% | 13.0% | 34.0% |
| | Female | 6 | 17 | 16 | 20 | 22 | 20 | 101 |
| | | 28.6% | 63.0% | 59.3% | 87.0% | 68.8% | 87.0% | 66.0% |
| Job title [n = 151] | Call handler/ customer services advisor | 3 | 8 | 12 | 9 | 26 | 19 | 77 |
| | | 14.3% | 29.6% | 44.4% | 42.9% | 81.3% | 82.6% | 51.0% |
| | Supervisor/ manager/team leader | 8 | 10 | 11 | 2 | 3 | 3 | 37 |
| | | 38.1% | 37.0% | 40.7% | 9.5% | 9.4% | 13.0% | 24.5% |
| | Other | 10 | 9 | 4 | 10 | 3 | 1 | 37 |
| | | 47.6% | 33.3% | 14.8% | 47.6% | 9.4% | 4.3% | 24.5% |
| Employment type [n = 151] | Full time | 21 | 23 | 25 | 7 | 23 | 12 | 111 |
| | | 100.0% | 88.5% | 96.2% | 31.8% | 69.7% | 52.2% | 73.5% |
| | Part time | 0 | 3 | 1 | 15 | 10 | 11 | 40 |
| | | 0.0% | 11.5% | 3.8% | 68.2% | 30.3% | 47.8% | 26.5% |
| How long have you been working for this contact centre? [n = 154] | <3 months | 0 | 0 | 0 | 0 | 3 | 4 | 7 |
| | | 0.0% | 0.0% | 0.0% | 0.0% | 9.1% | 17.4% | 4.5% |
| | 3–6 months | 0 | 0 | 0 | 1 | 3 | 3 | 7 |
| | | 0.0% | 0.0% | 0.0% | 4.3% | 9.1% | 13.0% | 4.5% |
| | 6–12 months | 0 | 0 | 6 | 7 | 0 | 1 | 14 |
| | | 0.0% | 0.0% | 22.2% | 30.4% | 0.0% | 4.3% | 9.1% |
| | 1–2 years | 7 | 1 | 4 | 1 | 3 | 3 | 19 |
| | | 33.3% | 3.7% | 14.8% | 4.3% | 9.1% | 13.0% | 12.3% |
| | 2–3 years | 2 | 1 | 2 | 3 | 10 | 2 | 20 |
| | | 9.5% | 3.7% | 7.4% | 13.0% | 30.3% | 8.7% | 13.0% |
| | >3 years | 12 | 25 | 15 | 11 | 14 | 10 | 87 |
| | | 57.1% | 92.6% | 55.6% | 47.8% | 42.4% | 43.5% | 56.5% |
| Health problems reported that may prevent participant from standing and moving more at work [n = 154] | No | 17 | 22 | 25 | 19 | 30 | 22 | 135 |
| | | 81.0% | 78.6% | 92.6% | 82.6% | 90.9% | 95.7% | 87.1% |
| | Yes | 4 | 6 | 2 | 4 | 3 | 1 | 20 |
| | | 19.0% | 21.4% | 7.4% | 17.4% | 9.1% | 4.3% | 12.9% |

Adapted from SUH NIHR report [29]

**Table 5. Baseline demographics for participants involved in post-lockdown data collection, stratified by centre.**

| | | Centre | | | | | Total |
|---|---|---|---|---|---|---|---|
| | | 3 | 6 | 7 | 10 | 11 | |
| **Total number of participants** | | 6 | 7 | 20 | 18 | 3 | 54 |
| Age (years) [n = 51] | 18–24 | 0 | 0 | 3 | 1 | 0 | 4 |
| | | 0.0% | 0.0% | 16.7% | 5.6% | 0.0% | 7.8% |
| | 25–34 | 2 | 1 | 11 | 2 | 2 | 18 |
| | | 40.0% | 14.3% | 61.1% | 11.1% | 66.7% | 35.3% |
| | 35–44 | 2 | 4 | 2 | 7 | 0 | 15 |
| | | 40.0% | 57.1% | 11.1% | 38.9% | 0.0% | 29.4% |
| | 45–54 | 1 | 1 | 1 | 5 | 0 | 8 |
| | | 20.0% | 14.3% | 5.6% | 27.8% | 0.0% | 15.7% |
| | 55–64 | 0 | 1 | 1 | 2 | 1 | 5 |
| | | 0.0% | 14.3% | 5.6% | 11.1% | 33.3% | 9.8% |
| | > 65 | 0 | 0 | 0 | 1 | 0 | 1 |
| | | 0.0% | 0.0% | 0.0% | 5.6% | 0.0% | 2.0% |
| Gender [n = 51] | Male | 2 | 2 | 13 | 6 | 0 | 23 |
| | | 40.0% | 28.6% | 72.2% | 33.3% | 0.0% | 45.1% |
| | Female | 3 | 5 | 5 | 12 | 3 | 28 |
| | | 60.0% | 71.4% | 27.8% | 66.7% | 100.0% | 54.9% |
| Job title [n = 51] | Call handler/customer services advisor | 3 | 2 | 18 | 18 | 3 | 44 |
| | | 60.0% | 28.6% | 100.0% | 100.0% | 100.0% | 86.3% |
| | Supervisor/manager/team leader | 2 | 3 | 0 | 0 | 0 | 5 |
| | | 40.0% | 42.9% | 0.0% | 0.0% | 0.0% | 9.8% |
| | Other | 0 | 2 | 0 | 0 | 0 | 2 |
| | | 0.0% | 28.6% | 0.0% | 0.0% | 0.0% | 3.9% |
| Employment type [n = 51] | Full time | 5 | 4 | 17 | 12 | 1 | 39 |
| | | 100.0% | 57.1% | 94.4% | 66.7% | 33.3% | 76.5% |
| | Part time | 0 | 3 | 1 | 6 | 2 | 12 |
| | | 0.0% | 42.9% | 5.6% | 33.3% | 66.7% | 23.5% |
| How long have you been working for this contact centre? [n = 51] | <3 months | 0 | 0 | 0 | 0 | 0 | 0 |
| | | 0.0% | 0.0% | 0.0% | 0.0% | 0.0% | 0.0% |
| | 3–6 months | 0 | 0 | 2 | 0 | 0 | 2 |
| | | 0.0% | 0.0% | 11.1% | 0.0% | 0.0% | 3.9% |
| | 6–12 months | 4 | 0 | 0 | 1 | 1 | 6 |
| | | 80.0% | 0.0% | 0.0% | 5.6% | 33.3% | 11.8% |
| | 1–2 years | 0 | 0 | 5 | 3 | 1 | 9 |
| | | 0.0% | 0.0% | 27.8% | 16.7% | 33.3% | 17.6% |
| | 2–3 years | 0 | 1 | 1 | 4 | 0 | 6 |
| | | 0.0% | 14.3% | 5.6% | 22.2% | 0.0% | 11.8% |
| | >3 years | 1 | 6 | 10 | 10 | 1 | 28 |
| | | 20.0% | 85.7% | 55.6% | 55.6% | 33.3% | 54.9% |
| Health problems reported that may prevent participant from standing and moving more at work [n = 154] | No | 4 | 7 | 14 | 16 | 3 | 44 |
| | | 80.0% | 100.0% | 77.8% | 88.9% | 100.0% | 86.3% |
| | Yes | 1 | 0 | 4 | 2 | 0 | 7 |
| | | 20.0% | 0.0% | 22.2% | 11.1% | 0.0% | 13.7% |

None of the participants previously worked for a company which used the SUH intervention. Adapted from SUH NIHR report [29].

**Table 6. Descriptive statistics of outcome measures.**

| | Time point | N | Missing | Mean | Median | SD | Min | Max | Q1 | Q3 |
|---|---|---|---|---|---|---|---|---|---|---|
| activPAL™: Sedentary time per day in the workplace (minutes) | A (Pre) | 28 | 1 | 367.5 | 365.8 | 103.6 | 117.8 | 541.1 | 288.1 | 441.6 |
| | B (Pre) | 43 | 1 | 379.5 | 386.2 | 103.1 | 158.7 | 659.2 | 321.2 | 440.7 |
| | C (Pre) | 23 | 0 | 322.5 | 325.5 | 89.7 | 134.0 | 461.6 | 256.0 | 383.2 |
| activPAL™: Sedentary time per day while awake (minutes) | A (Pre) | 28 | 1 | 676.0 | 660.2 | 108.8 | 419.0 | 910.0 | 609.7 | 767.9 |
| | B (Pre) | 41 | 3 | 658.8 | 651.6 | 127.6 | 334.4 | 892.7 | 600.8 | 753.1 |
| | C (Pre) | 22 | 1 | 612.9 | 602.6 | 90.8 | 483.4 | 783.3 | 523.2 | 675.3 |
| OSPAQ: Hours sitting at work per week | A (Pre) | 49 | 2 | 32.3 | 29.6 | 15.5 | 7.2 | 72.0 | 21.0 | 42.0 |
| | B (Pre) | 60 | 0 | 26.6 | 27.0 | 8.5 | 4.8 | 39.2 | 21.6 | 33.6 |
| | C (Pre) | 43 | 1 | 24.6 | 25.2 | 7.5 | 6.0 | 37.2 | 21.0 | 31.2 |
| | D (Post) | 25 | 0 | 29.0 | 30.4 | 10.2 | 5.0 | 49.5 | 24.4 | 35.3 |
| | E (Post) | 25 | 0 | 29.4 | 30.4 | 10.6 | 7.2 | 47.0 | 21.4 | 36.9 |
| OSPAQ: Minutes sitting at work per day | A (Pre) | 49 | 2 | 417.5 | 421.1 | 180.4 | 96.0 | 720.0 | 265.2 | 576.0 |
| | B (Pre) | 60 | 0 | 343.0 | 360.0 | 96.0 | 135.0 | 540.0 | 266.4 | 422.1 |
| | C (Pre) | 43 | 1 | 330.1 | 329.3 | 62.9 | 154.0 | 446.0 | 294.0 | 378.0 |
| | D (Post) | 24 | 1 | 379.5 | 368.2 | 101.7 | 60.00 | 594.0 | 338.6 | 445.2 |
| | E (Post) | 23 | 2 | 394.2 | 403.2 | 78.6 | 180.0 | 552.0 | 352.4 | 432.0 |
| WEMWBS Total Score | A (Pre) | 50 | 1 | 47.8 | 48.0 | 8.3 | 30.0 | 70.0 | 42.0 | 53.3 |
| | B (Pre) | 60 | 0 | 49.1 | 50.5 | 7.5 | 28.0 | 64.0 | 44.0 | 54.0 |
| | C (Pre) | 44 | 0 | 49.5 | 50.0 | 7.3 | 34.0 | 67.0 | 44.0 | 54.0 |
| | D (Post) | 25 | 0 | 44.0 | 44.0 | 6.8 | 32.0 | 56.0 | 38.5 | 50.5 |
| | E (Post) | 25 | 0 | 43.6 | 42.0 | 7.9 | 32.0 | 68.0 | 38.0 | 48.5 |
| UWES Total Score | A (Pre) | 50 | 1 | 4.1 | 4.2 | 1.0 | 1.4 | 6.2 | 3.6 | 4.8 |
| | B (Pre) | 60 | 0 | 4.9 | 5.0 | 0.9 | 2.4 | 6.7 | 4.4 | 5.6 |
| | C (Pre) | 43 | 1 | 4.8 | 4.7 | 0.9 | 2.5 | 6.5 | 4.4 | 5.6 |
| | D (Post) | 25 | 0 | 4.6 | 4.7 | 0.7 | 2.9 | 5.7 | 4.2 | 5.0 |
| | E (Post) | 23 | 2 | 4.6 | 4.5 | 1.0 | 2.5 | 6.8 | 4.2 | 5.1 |
| MSK-HQ Total score | A (Pre) | 42 | 9 | 28.1 | 25.0 | 12.0 | 14.0 | 55.0 | 17.0 | 40.3 |
| | B (Pre) | 58 | 2 | 25.5 | 23.5 | 10.6 | 14.0 | 61.0 | 17.0 | 28.0 |
| | C (Pre) | 35 | 9 | 25.9 | 26.0 | 10.9 | 14.0 | 53.0 | 14.0 | 16.0 |
| | D (Post) | 25 | 0 | 27.0 | 26.0 | 9.1 | 14.0 | 49.0 | 19.0 | 34.0 |
| | E (Post) | 23 | 2 | 24.8 | 25.0 | 6.8 | 15.0 | 41.0 | 19.0 | 30.0 |

Adapted from SUH NIHR report [29]. Refer Table 3 for details on time points

**Progression criteria.** A list of five progression criteria were pre-specified, for determining whether we could progress to a larger randomised control trial in future [see Parker et al. (2020) [30]]. Our evaluation of each of these criteria was published previously in the full report [29], and is also provided in a S4 File. Three out of five key progression criteria were achieved with good recruitment rates and retention. One criterion was partially achieved and one was not achieved. The progression criterion that was not achieved was the inclusion of a 45 minutes reduction in primary outcome in our 95% confidence interval. However, the reliability of this analysis was questionable due to the impact of the COVID-19 pandemic on the study design, rendering the analysis results highly vulnerable to bias.

## Process evaluation results

Thirty-three staff and managers from six centres participated in the process evaluation focus groups and interviews. Four focus groups (22 participants) and three interviews were

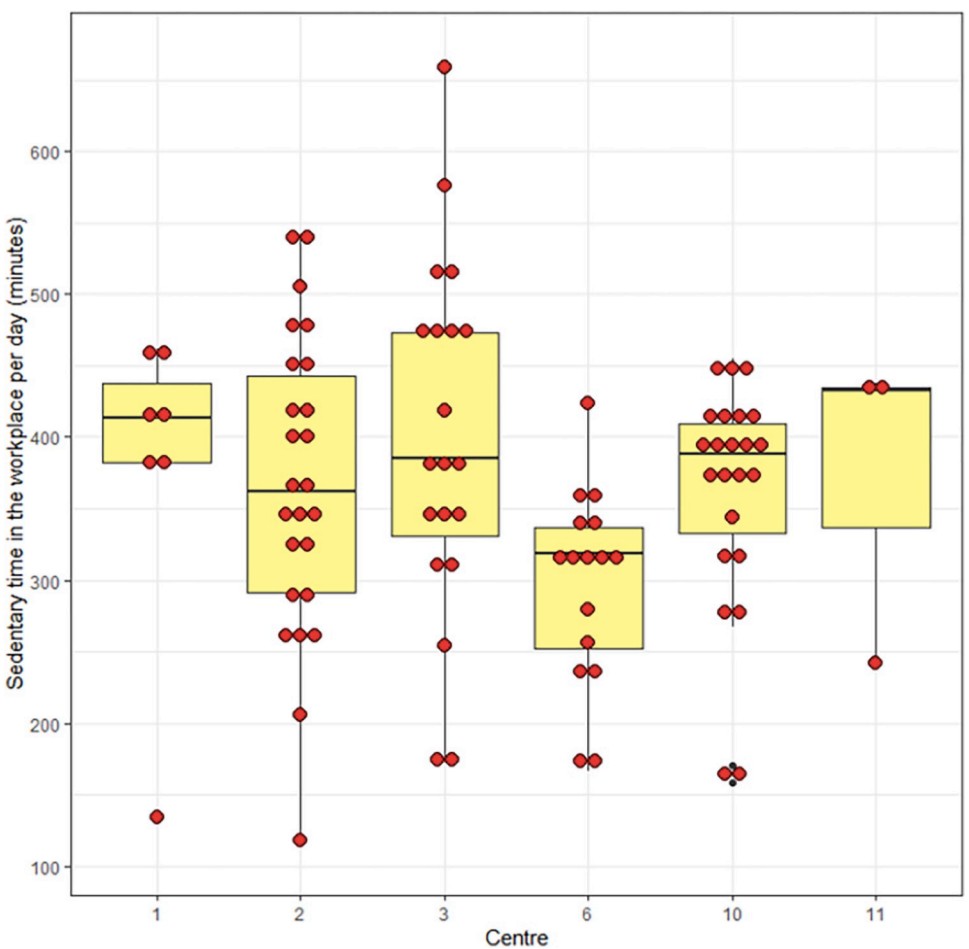

**Fig 3. Box and dot plot of primary outcome stratified by centre.**

conducted with staff members. Interviews were conducted with managers from the six centres (eight participants). Managers interviewed included Head of Customer Service, Customer Service Coordinator, Contact Centre Manager, staff engagement and wellbeing managers, HR Advisor, and Centre Union Rep. Activity preferences are reported from 51 staff who completed the post-intervention questionnaire for the pre-lockdown period, and 45 participants who completed the post-intervention questionnaire post lockdown.

**Reach: Programme significance and appeal.** The majority of staff described participating in the pre-lockdown SUH intervention as a positive experience overall. It was felt that SUH was a particularly significant and unique programme as it encouraged movement in a sedentary environment where staff are tied to the desk, and also brought attention to the lack of movement in this environment. The importance of the SUH programme in improving physical, mental, emotional and social wellbeing was emphasised by both staff and managers. Managers expressed that the SUH programme helped them look after their staff better, and staff associated SUH with health and wellbeing in the workplace, targeting musculoskeletal issues and mental health. Encouraging team-work, uplifting team spirt, and providing a morale boost were aspects valued by staff and managers.

'*I honestly think it was a really positive experience. I think, for some people, it made a difference. It really kind of. . .it changed the way they worked. . .as an employer, giving our staff the*

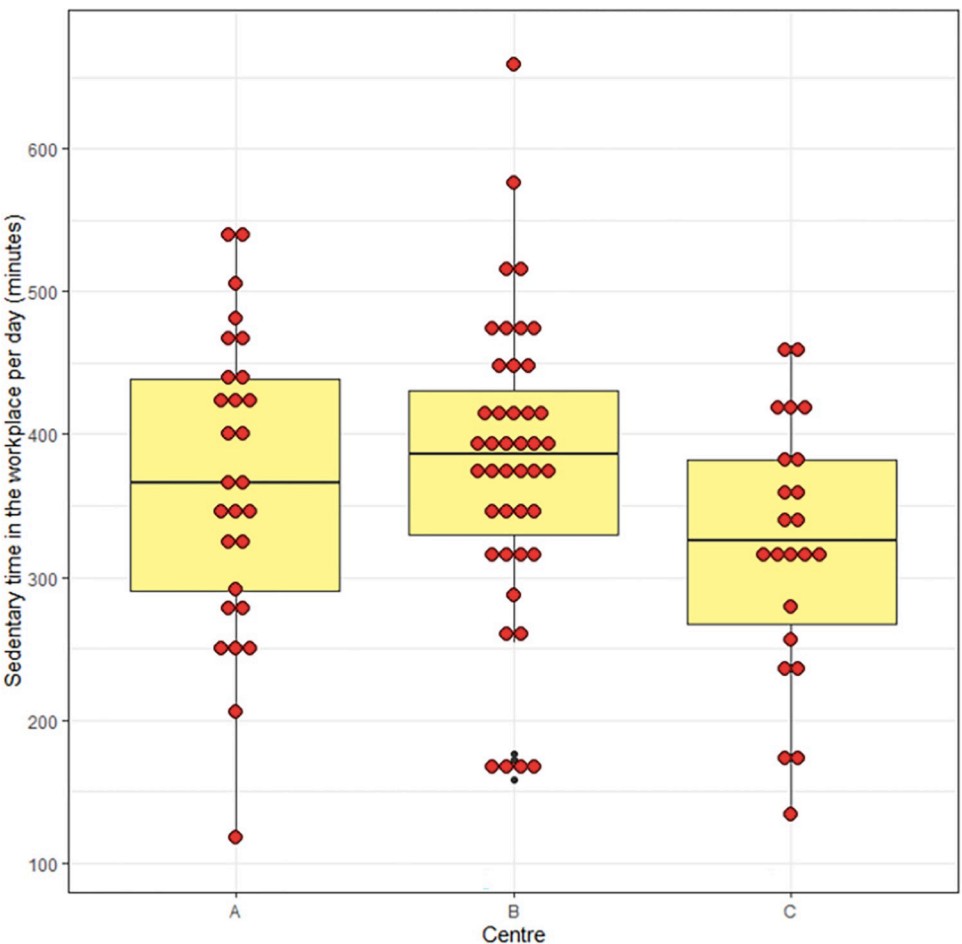

**Fig 4. Box and dot plot of primary outcome stratified by data collection group.**

*opportunity to work differently, is fantastically positive. You can't force people to do it, but the fact that we're able to give them the opportunity to work differently, to move around and, you know, to look after themselves and keep well, while they're working, I think, is fantastic. I think it's just been brilliant.' (Centre 10 Manager)*

The programme continued to be valued in the post-lockdown period, with staff associating SUH with looking after themselves which was considered especially important during lockdown. During a tough period, where staff were having feelings of depression, and did not really take time for themselves, staff felt that SUH provided a purpose and a variety of activities to help them cope. The post-lockdown activities provided staff with motivation and reminded them why it is important to sit less and look out for their wellbeing.

*'Everyone who works in an office job knows that pain when you try and get up in the morning and you've got that shooting pain in the back everyone feels that. It doesn't matter what job you do. So I think the message you guys project is so pure that people will just naturally rotate around it because literally you're doing it for the betterment of people.'*

*(Staff, Centre 7)*

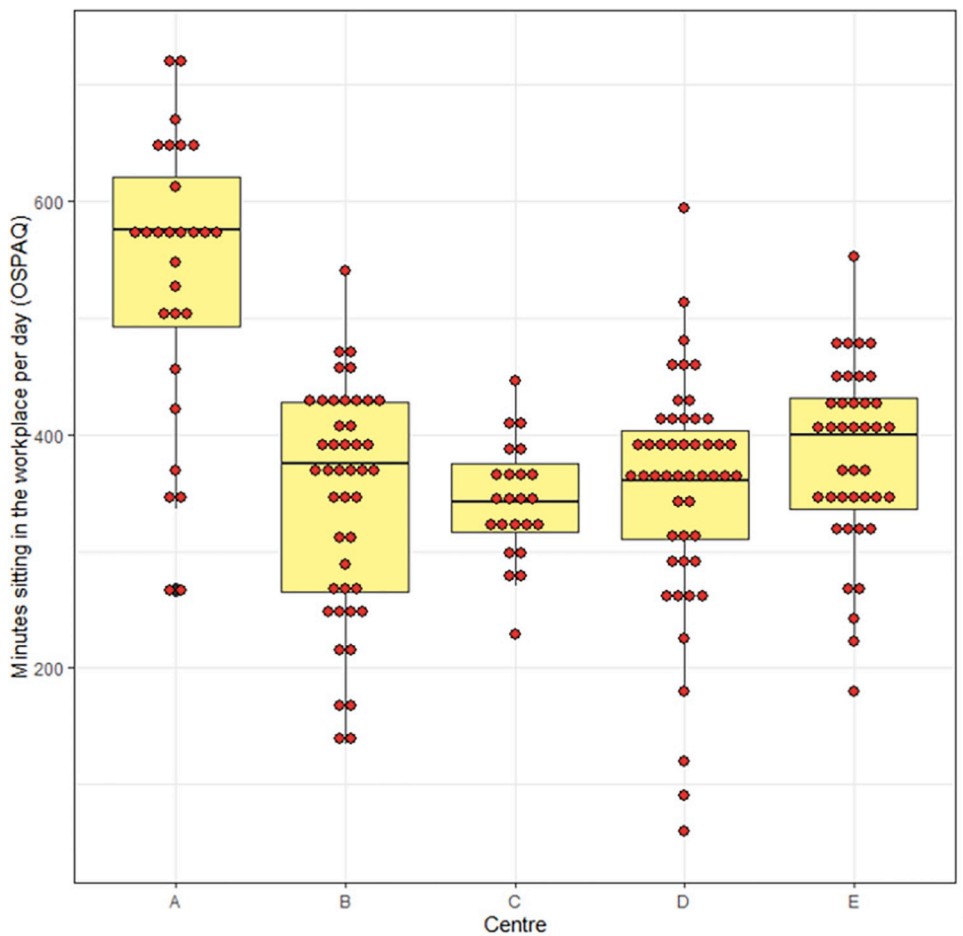

**Fig 5. Box and dot plot of self-reported sedentary time stratified by data collection group.**

The individual consultations provided a much-needed wellbeing 'check-in', and the resources helped staff mitigate increases in sedentary behaviour and decreases in physical activity due to the restrictions on activities during the pandemic and, for many, the shift to home working. During a time when social gatherings and activities were challenging, SUH activities such as the step count challenges provided a fun way to encourage and support staff to engage in physical activity, with competitive elements which many staff welcomed and enjoyed.

**Reach: Programme participation.**   All centres reported that more than 50% of staff participated in at least one aspect of SUH activities during the six-month pre-lockdown intervention period. Centres felt that several staff members were actively involved in the programme and participated in activities on a consistent basis. Sixty-five percent of staff reported using SUH equipment or participating in activities once or more during the pre-lockdown period (Table 7).

During the post-lockdown period, working from home limited the ability of managers to oversee or check participation levels. It was hence difficult for managers to comment on participation in the SUH intervention. The SUH team struggled to maintain contact with staff working from home and make them aware of SUH activities. While the consultations were offered to all staff, several could not participate as they did not have access to MS Teams. In addition,

**Table 7. Use of equipment or participation in activities during the pre-lockdown SUH programme.**

| Activity | Proportion of staff that used equipment or participated in activities once or more during the SUH programme (n = 48) |
|---|---|
| Desk-based equipment (e.g. Treadmill, standing desk) | 31% |
| Non desk-based equipment (e.g. Powerplate, table tennis) | 15% |
| Mindfulness (e.g. LEGO, colouring, jigsaw) | 25% |
| Group activities (e.g. group walks, team challenges) | 13% |
| Individual activities (e.g. walks, running, cycling) | 38% |

Note: data not available for three participants, proportions are based on 48 participants. Adapted from SUH NIHR report [29]

SUH equipment was mainly available at offices, and to staff from some centres where equipment borrowing procedures for home working had been established. Of those who completed the post-lockdown questionnaire, 95% of staff participated in at least one activity (Table 8)

**Effectiveness: Perceived benefits.** The staff and managers from participating centres felt that the pre-lockdown SUH programme was successful with positive outcomes, and that the programme kindled a shift to healthier thinking and behaviour. They consistently noted an increase in physical activity and reductions in sedentary behaviour among staff. Several other perceived benefits were reported including:

Mental health benefits: It was felt that the SUH programme improved staff morale and mood. Staff reported feeling happier, more relaxed, energised and alert. Staff felt that SUH helped manage stress and cope with stressful calls. One manager reported an observed reduction in staff absence due to mental health issues since the SUH study started.

**Table 8. Participation in activities during the post-lockdown SUH programme.**

| Activities | Proportion of staff that participated in different activities during the post-lockdown SUH programme (n = 41) |
|---|---|
| Social activities (example: Step count challenge, walks/activities with those in your household/colleagues): | 44% |
| Virtual social activities with an active component (bingo/quiz with active component/virtual social exercise classes) | 59% |
| Individual activity–Goal setting | 54% |
| Individual activity–Desktop stretches | 51% |
| Individual activity–Exercise videos and apps | 35% |
| Individual activity–Other (e.g., walking, running, cycling) | 85% |
| Individual activity–Used mindfulness resources | 36% |
| Individual activity–Used Stand Up for Health website: | 10% |
| Environmental activity–Made changes to desk space to help reduce sitting | 37% |

Note: data not available for four participants, proportions are based on 41 participants. Adapted from SUH NIHR report [29].

Physical health benefits: Staff reported reductions in weight, improvements in musculoskeletal issues and health conditions such as Reynauds syndrome, as well as improvements in blood pressure.

*R2: I've actually gone down from 14 stone seven to 13 stone eight in the last couple of months. I'm really feeling better. My blood pressure's improved ad I just feel generally ten times better than I did a few months back.*

(*Staff, Centre 11*)

Staff engagement and productivity: Employees and managers perceived that, as a result of the SUH programme, staff were engaging as team and supporting each other. Managers felt staff were more efficient and alert and noted improvements to productivity as well as reduced sickness absence.

'*. . .It made a difference to sickness absence. It made a difference to productivity. So for us, kind of call time and after call time, before the programme, was 12 minutes. After the programme, it was about ten minutes.*'

(*Manager, Centre 10*)

Staff perceived several benefits from the post-lockdown programme including increased physical activity, improved mental and physical health, increased focus and concentration and reduced stress. Benefits were mainly mentioned by Centre 6 and 7 and related to the step count challenge and use of equipment.

'*I believe it was a balance board you guys had sent in so I really enjoyed the balance board.. . . but it was really good to sort of get that level of concentration because I feel like especially in my sort of job when you have a monotonous day, it's just one call after another, the same old people trying to save money, so the fact that I could go on my break and when I was on the balance board I had no choice but to fully focus. . .I think that was a good thing because it gets your attention off the long monotonous day that you have ahead of you. . .*'

(*Staff, Centre 7*)

**Adoption.** 34 centres from across the UK were targeted, and 16 among these got in touch with the SUH team expressing an interest to participate. Of those centres, 11 actually participated, as the study protocol required only 10 centres to take part.

**Implementation: Organisational change.** During the pre-lockdown period, most centres felt that the various levels of the organisation supported the programme, and SUH was seen as a part of the agenda to promote health and wellbeing among staff.

'*we've kind of given them something to think about in terms of us going for a health award in the workplace because they see Stand Up for Health as being a really key part of that, not just actually obviously within the customer service team but something that can be implemented across the whole organisation. . . I think Stand Up for Health is seen as part of a bigger picture which is about making sure we have happy, healthy staff.*'

(*Manager, Centre 11*)

Initiatives undertaken at the organisational level included the sanctioning of handsfree wireless headsets (Centre 11), allocating an extra 30-minute SUH break on a weekly basis (Centre 10), and the setting up of a SUH committee (sometimes an existing health and wellbeing committee) consisting of multiple staff members across teams in four centres (Centres 2, 11, 3, 10). These committees were perceived by participants as integral to successful implementation of the programme through developing, operationalising and implementing several activities. In one centre (Centre 10), they also promoted the SUH equipment, thinking through safety and arranging logistics (instructions, photos, hygiene, booking system).

Two centres did not report any initiatives implemented at the organisational level (Centres 6, 2), and the SUH committee set up in Centre 2 was not very active. These centres had more rigid organisational set ups, and implementing new ideas was perceived by managers as a time-consuming process. However, it was felt that the SUH programme had initiated change, and there were discussions around ideas and suggestions at the organisational level. These small steps were seen as important changes.

Only few organisational changes occurred as a result of the post-lockdown programme. In Centre 7, higher management were aware of the programme and discussed and supported SUH activity implementation. Centre 7 buildings were being refurbished, and there were plans to incorporate elements and principles of the programme to encourage more movement among staff (example: pitch and putt areas).

**Implementation: Environmental changes.** Staff described enjoying using the various pieces of physical equipment supplied by the research team as a part of the pre-lockdown programme such as a ball desk chair, desk bike, treadmill and standing desk risers (Table 9). Staff especially liked equipment that created social interaction and competitions such as table

**Table 9. Equipment and activities that contact centre staff used or participated in during the pre-lockdown SUH programme.**

| Popular equipment | Mindfulness equipment/ activities | Social activities | Individual activities |
|---|---|---|---|
| Ball chair<br>Vibration plate<br>Stepper<br>Peddles<br>Exercise bands<br>Wrist weights<br>Power spin<br>Table tennis<br>Treadmill, cycle<br>Bike<br>Dart board<br>Standing desks/Sit-Stand. com desk-risers | Colouring in<br>Jigsaw<br>LEGO<br>Wordsearch | Pokemon group<br>Dungeons and dragons<br>Dancing<br>Stretching<br>Book club<br>Table tennis<br>Exercise Lucky dip Friday<br>6-week health challenge (food related)<br>Weekly Clubercise class with glow sticks<br>Cocktails<br>Jigsaw challenge<br>Quiz<br>Parkrun<br>Mexican wave across desks<br>Bingo<br>Team target- stand for sometime and pass it on<br>Stepper challenge<br>Lunch time walks<br>Organising activities outside work<br>Football teams<br>Exercise sessions on the call floor<br>Gaming session after work | Colouring in<br>Biking<br>Walk during breaks<br>Goal setting- getting an activity tracker and challenging themselves to 10,000 steps<br>Swimming<br>Word search<br>Stretching when they stand around<br>Reminders on computer to take stand up/stretch breaks<br>Desktop stretches<br>Jigsaws |

Adapted from SUH NIHR report [29]

tennis, darts, peddles, and the stationary bike, as well as equipment they could use at their desk such as exercise bands, wrist weights, and power spin. Staff who had limited break length and frequency preferred activities which could be done at the desk, such as colouring, and the use of handheld equipment such as bands, as well as under desk pedals and ball chairs. They enjoyed mindfulness activities like colouring, jigsaw, LEGO, wordsearch and felt that these simple activities made a big impact. These non-physical activities were appealing to those who did not want to take part in physical activities, and some mindfulness activities such as colouring and jigsaws became a team effort. However, some staff reported that they found the colouring and LEGO childish. Initiatives at the environmental level in the participating centres included:

- Meetings held in a room where equipment was set up, with the staff using different pieces of equipment (Centre 10)

- Setting up a dedicated room for exercise classes with yoga mats, wrist weights, hand weights, purchased disco lights (Centre 10)

- Purchased handsfree wireless sets after a successful trial (Centre 11).

- Based on the SUH experience, centres purchased several pieces of equipment (Centre 11: two or three ball chairs, cycle equipment, colouring pens, jigsaws, LEGO; Centre 10: small weights).

- Staff also brought in their own equipment (stepper, twisting discs, peddles, hula hoops, colouring, word searches, puzzles, jigsaws in Centres 10, 11, 3)

- Set up reminders on the computer to take a break (Centre 11).

- Centres came up with mindfulness initiatives (Centre 10: purchased jigsaws and supplied more colouring sheets; Centre 2: planned mindfulness sessions)

With a large proportion of staff working from home, staff felt that work environments were quite constrained during the post-lockdown period. Staff made small changes such as moving around the house during breaks and using the toilets upstairs to increase activity. They also made some ergonomic changes including purchasing a more comfortable chair and using a yoga ball chair. Staff enjoyed using equipment (balance boards and ball chairs), either in the office (if they were able to go in) or at home (if their centre arranged for them to sign out items).

**Implementation: Social change.**   Staff and managers felt that the social activities implemented (Table 9) helped bring groups together, support each other and cultivate new friendships. Through sharing stories and ideas, and promoting teamwork and engagement, the programme generated motivation and enthusiasm to stand up more and be active. Staff from one centre mentioned increased interaction during later shifts. SUH equipment, mindfulness activities and competitions provided opportunities for social interaction, and the range of activities engaged more shy staff members, as well as those who were not initially interested in the programme.

'..*At the beginning, me personally, I was like, oh, I'm so excited—[the vibroplate] and the treadmill. But to be honest, it was more of the engagement activities, it was more of the ping pong and things like that, that you were doing with other people that, you know, got me involved. And I think that's what a lot came out of. . .with it was that the engagement that it made across the team as well, it just heightened everything, it was really good*' (Staff, Centre 10)

Staff and managers from two centres (Centres 2 and 6) reported that they were not aware of any social activities being implemented as a part of SUH. In one centre, logistical issues (e.g., lack of space/time) impacted on the implementation of some activities despite manager buy-in and enthusiasm from staff.

During the post-lockdown period, it was felt that SUH generated social interaction through using the equipment and competitions at the workplace, but it was more challenging while working from home. The Stepcount challenge prompted staff to communicate with each other on WhatsApp, and inspire and motivate each other. SUH also enabled staff to get more active with family members. Another effect of SUH was that when lockdown was eased, staff were meeting each other and walking in parks and at the beach.

**Implementation: Individual change.** Staff members described individual level activities that they undertook (Table 9). They reported feeling motivated to take up activities, and participated in activities to make themselves feel better, even feeling inspired to do things outside the office. Seeing other people get up and do things reminded and prompted them to get involved with activities and be more active. Another mechanism was goal setting (example—celebrating when they achieve their goal of 10,000 steps a day).

The motivation to increase activity during the post-lockdown period came mainly from the Stepcount challenge, consultations and the activity plan. Some staff purchased an activity tracker to help reach the target step count. The Stepcount challenge helped create routines that enabled an increase in physical activity in the longer term. In addition, staff were motivated to do exercises as they helped with musculoskeletal issues such as a stiff neck. The manager from Centre 7 noticed a change in thinking among staff and a shift to healthier and more active behaviour (staff may take a walk rather than sit and browse on their mobile phones). Motivation to move more during breaks and finding ways to increase activity also stemmed from a desire to improve mental and physical health. Staff expressed that the consultation sessions and activity plans provided tailored suggestions to improve mental health and increase physical activity.

**Implementation: Ownership.** A sense of ownership among staff is a proposed mechanism of change within the SUH programme theory. In most centres, staff described feeling a strong sense of ownership. They were very involved and engaged in the programme and suggested activities (example: Pokemon Go, Dungeons and Dragons, Clubercise class, walking, see Table 9), brought in their own equipment and supplies (example: funny colouring in, colour pencils, jigsaws, peddles). Some staff also acted as role models that the teams would follow.

I: *And which aspect of the programme do you think worked well*?

R: *I think the actual getting people involved in coming forward with their own activities of what we could do, because I think that, sort of, went out to the wider audience. . .*

*(Manager, Centre 3)*

However, in two centres (Centres 2 and 6), staff did not feel a sense of ownership. They felt that voicing an opinion did not lead to implementation or change in their centre due to logistical barriers (venue, time, public liability insurance). They noted some progress towards a more inclusive approach, with discussions and meeting being held to procure suggestions from teams.

Feelings of ownership were not strong during the post-lockdown programme, as staff were more isolated while working from home. Some staff reported being able to make suggestions and contribute to the step count challenge.

**Implementation: Sedentary behaviour and physical activity awareness.** There was an increased awareness of the importance of sitting less and moving more in contact centres. Staff

were also more aware of the risks associated with prolonged sitting, as well as the positive aspects of sitting less and moving more. Some staff noted that they had not thought about sedentary behaviour before the programme, and felt that SUH had highlighted the dangers of being sedentary especially in a contact centre environment. They noted that SUH had spurred staff on to look for solutions such as wearing activity trackers, searching for equipment on online retail outlets like Amazon, and taking more breaks. Staff from one centre (Centre 10) also felt that participating in SUH made them realise that you don't have to stop work to be active. In this centre, the manager reported that the increased awareness among staff about sedentary behaviour and physical activity had longer term effects and helped staff stay active and sit less during the transition to homeworking.

*R3*: *I would agree with Name. I think for me I had recognised that I sit too much during work and there's been so much in the media now, we know that it's not good for you. I think it just means that you're trying to improve your mental wellbeing as well as your physical wellbeing.*

*R5*: *All the mindsets are being changed since we started this, just even little things like sitting at your desk doing a puzzle, you're not just sitting there, you're doing something. Everybody's changed in a way, even if it's just a little way.*

*I1*: *So, you think the mindset's changed as a result of Stand Up for Health?*

*R5*: *Uh Huh, yeah.*

*(Staff, Centre 11)*

During the post-lockdown programme, staff felt that the consultations helped create awareness about SB and PA, and encouraged them to think about moving more and about physical activity guidelines. They noted that the consultation sessions made them aware that they were even more sedentary while working from home. A manager (Centre 7) perceived that SUH was instrumental in initiating a change to more healthy and active behaviour among staff.

**Implementation: Unintended consequences.**   Three positive unintended consequences of the pre-lockdown programme were reported by staff. The first was the adoption of a healthier diet by staff members, both individually and at group work events. The second was increased physical activity outside of work including swimming and Parkrun (free, community event where you can walk, jog, run, or volunteer). These were reported by several staff members and managers from Centre 11. Finally, there was interest and participation from other centres, departments and teams. Staff and managers from Centres 2, 11 and 10 mentioned that those visiting from other centres were intrigued by the equipment, and even used some of the equipment and participated in activities that were organised as a part of SUH. A negative unintended consequence mentioned during the post-lockdown period by one staff member from Centre 7 was the potential for injury while using equipment such as a balance board. A positive unintended consequence of the post-lockdown programme was the adoption of healthier diets by staff reported by two staff members from Centre 7.

**Maintenance.**   A key aspect influencing the maintenance of implementing the SUH intervention is having an appropriate budget, primarily for physical equipment. Several centres reported having budget approval, based on the creation of a business case around improvement in productivity (call time) and sickness absence as a result of programme feedback. This has, or will, lead to purchase of equipment such as wireless headsets, desk bikes, desk raisers and smaller items such as small weight sets and yoga mats. However, it was noted that purchasing of equipment in some centres had been put on hold due to COVID-19.

The centres appreciated that the programme was adapted for the COVID-19 pandemic lockdown. Staff and managers expressed that they would like to continue and focus on SUH after the pandemic. Managers acknowledged that creating change was a slow process, and expected the programme to grow slowly, with benefits accruing in the long run. Some practices (huddles, walking) and initiatives (wireless headsets) have already been set up and will continue. The programme has created an awareness of sedentary behaviour and sparked numerous ideas among staff for working on site as well as at home.

## Discussion

This is a novel study that addresses the need for multi-component, context specific programmes targeting sedentary behaviour in contact centres, and adds to the limited literature in this area. The SUH programme was developed though a rigorous process as an adaptive intervention that considers organisational, environmental, social, and individual factors [22]. Although the project was hampered by the COVID-19 pandemic and the ensuing lockdown, the study provides valuable insights relating to intervention implementation and acceptability. The SUH programme was found to be feasible in most centres, and several activities aligning with each level of the programme's theory of change were implemented. The programme was valued as an initiative that improves the health and wellbeing of contact centre employees, and numerous physical and mental health benefits were reported by staff and managers. In addition, four out of five key progression criteria were achieved (at least partially), with good recruitment rates and retention.

### Summary of findings in relation to research objectives

In this paper, aligned with research objective 1, we have reported on the variability of our primary and secondary outcomes, which will help when planning future larger studies. The OSPAQ showed very high variability in our study which may rule out this outcome as a primary outcome measure in future trials. In contrast, the ActivPAL-based physical activity outcome, exhibited lower variability and so may have greater power to detect real changes between intervention/control conditions in a future trial. However, ActivPAL-based measures place a greater burden on participants [43], so recruitment or retention rates may be compromised. We will explore and report on the acceptability and feasibility of data collection instruments and processes elsewhere, taking into account qualitative and quantitative data.

With respect to research objective 2, the pre-lockdown SUH programme was well received by the centres. Contact centre staff and managers reported several benefits including a perceived increase in physical activity, reduced sitting, improved morale and mood, and a reduction in stress. Most centres reported that staff members participated actively in the programme, and more than 50% of staff participated in at least one SUH activity. In most RCT and feasibility studies, those who participate in the intervention also complete the assessments. For example, a feasibility study in contact centres recruited six of 20 team leaders and 17 of 84 call agents (25 eligible) to participate in the intervention and feasibility assessments [16]. These figures cannot be compared with participation in the SUH programme where the whole centre was exposed to the intervention, reflecting a more "real world" implementation, with a proportion participating in the evaluation. The post-lockdown programme was valued by staff and managers, and welcomed as a timely intervention to promote wellbeing and physical activity and target sedentary behaviour. However, the programme was adapted and delivered at a time where organisations were setting up processes for working from home, and there were restrictions to in-office working. Hence there were several constraints in terms of what could be delivered, as well as reaching staff members who were working from home.

To address research objective 3, we investigated the centres' fidelity to the theory of change. Contact centre staff implemented several initiatives at the organisational, environmental, social and individual levels. Staff in many centres also expressed a strong sense of ownership during the pre-lockdown programme. Creating a sense of ownership through empowerment and shared decision making is an efficacious intervention component that has resulted in positive behaviour change and outcomes [44–46]. SUH aimed to instil a sense of ownership among employees through opportunities to express activity preferences and propose ideas during the workshops, and setting up of a SUH committee. An increased awareness of the significance of sitting less and moving more was reported by contact centre staff, and is a mechanism cited by other contact centre interventions [16]. Researchers worked with the centre managers to choose activities (aligned to the programme theory of change) that would work for the centre, and fidelity to the intervention is enhanced due to the non-prescriptive nature of the intervention. However, two centres did not adopt any initiatives at the organisational level and staff did not express that they felt a sense of ownership. This ties in with differences across centres and is explored further while discussing objective 4. During the post-lockdown period, few or no initiatives were implemented at the organisational and environmental levels. Staff reported that the post-lockdown programme created awareness about sedentary behaviour and physical activity, but they did not feel a strong sense of ownership over the programme. Unintended positive consequences included the adoption of a healthier diet, increased physical activity outside work, and interest in the programme from other centres and teams. This highlights the potential of SUH to impact on health and wellbeing outcomes beyond sedentary behaviour. While no injury was reported, the risk of injury was mentioned by participants as a possible negative consequence.

The final research objective aimed to understand differences in programme delivery between centres. Contact centres have complex and varied environments and systems which can influence programme delivery and success. As mentioned earlier, some centres did not see organisational change and staff members did not experience a sense of ownership. Reasons for these differences in fidelity to the theory of change could be attributed to:

1. Communication: Effective communication and coordination between the SUH team and centres, as well as between individuals and teams within centres, is important and can impact on staff engagement, and provision and usage of SUH equipment. In addition, enthusiasm and support from managers can ensure that staff are aware of the programme and feel like they are encouraged to participate. Managers being hands-on with the programme, and promoting it among staff (Centres 11, 10) enabled successful implementation. In one centre (Centre 6), poor communication led to removal of equipment due to safety concerns. Communication has been identified as an important aspect that influences participant engagement and impacts programme success [47].

2. SUH committee: Setting up a SUH committee generated ownership among staff, encouraged ideas, as well as aided the implementation of activities. The committee also managed logistical aspects such as setting up a rota for standing desks and hygiene measures for the equipment. Centres where the SUH committee was set up and was active saw more successful implementation.

3. Organisational support: Change at the organisational level, in particular organisational culture can be difficult to create [48]. It is particularly challenging to implement programmes and create organisational change in centres that are more bureaucratic, rigid and resistant to change such as those based within public organisations [49].

### Strength and limitations

**Strengths.** This study has strengths relating to the SUH intervention and evaluative methods. Using a multicomponent approach based on the socioecological model provided a strong foundation for the programme, allowing for ease of programme activity idea generation and implementation. The programme was developed using an intervention development framework [25, 26], which also involved the creation of a theory of change model. The adaptive, non-prescriptive nature of the intervention meant that activities and initiatives could be tailored and implemented within the unique context of each contact centre. Contact centre employees were involved in deciding on the intervention activities, creating ownership [44–46]. The variety of activities introduced meant that there were options to cater to diverse interest and abilities.

The study included both a preliminary outcome evaluation and a process evaluation, and used a mixed method approach to address the research objectives [28]. A variety of data collection methods were used including device-based measures to assess sedentary behaviour and self-report measures for other outcomes. The process evaluation used the RE-AIM framework to develop the topic guide and analysis plan, allowing for a robust evaluation [32, 33]. Interviews and focus groups were conducted with a substantial number of staff and managers across the contact centres to procure in-depth accounts of the experience of the intervention. Finally, while other trials and feasibility studies on sedentary behaviour in contact centres have been conducted with a single centre [16–18], this is the only study to our knowledge to have included multiple contact centres across the UK and was able to draw out aspects impacting implementation across contact centres.

**Limitations.** A limitation of the pre-lockdown intervention is that it is time intensive for the SUH team as well as the contact centre, requiring planning, coordination, communication and carrying out the implementation activities. It was challenging for staff in some centres to partake in non-desk based SUH activities as their shifts were quite rigid and they had little flexibility for taking time away from their desks.

Since participation in data collection is voluntary, another limitation is that the evaluation may predominantly capture outcomes and views from enthusiastic and interested staff. A number of centres were engaged in the North East Better Health at Work Award, implying that the sample may have a bias in including centres that are actively considering health improvement activities.

The COVID-19 pandemic compelled researchers to abandon or change their research plans [50], and the SUH study was also severely impacted. The associated lockdowns disrupted the research design and programme delivery, leading to incomplete data collection, and we could not fully implement our original statistical analysis plan. This has also limited the generalisability of study results.

### Recommendations

During the process evaluation, we compiled a list of suggestions (Box 1), which should feed into refining the programme for future studies. These recommendations will be useful for all research relating to contact centres. While the programme was delivered successfully in several centres, there were barriers to implementing activities at the organisational level in some cases. Future studies should aim to understand organisational level constraints and facilitators, and investigate ways to influence organisational factors to aid intervention delivery. The post-lockdown intervention was developed and implemented rapidly, and would need to be developed further, taking into account the findings from the process evaluation, and recent studies on hybrid working.

### Box 1. Key recommendations for programme adaptation

Ensure appropriate placement and labelling of, and communication about, equipment for staff.

- Incorporate or suggest more competition-type activities for staff within and between teams and organisations, and showcase this using the programme website.

- Share results and benefits of the programme, especially at the individual level, to motivate staff (example- sharing individual level sedentary behaviour data).

- Have a higher amount and variety of communication and advertising measures to create awareness of the programme and check in with staff. This advertising and communication should include information relating to the risks of high rates of sedentary behaviour and low levels of physical activity to emphasise the programme's importance and increase motivation to participate.

- Visit the centres more often and at different time points to maximise in-person contact with staff.

- Encourage managers and others at various management levels to actively support the programme and participate in programme activities.

- Set more realistic expectations with respect to the speed at which implementation of activities, and changes to behaviour and outcomes (at the various levels) are realised.

Despite being impacted by the COVID-19 pandemic, the SUH programme shows potential as an appealing and acceptable intervention with several perceived benefits. Future studies should take into account a staff turnover rate of approximately 20% over a 10–12 month period. There were no participants who stated that they had previously worked for a company which had used the SUH intervention, even among the post-lockdown participants, which suggests that contamination between intervention and control groups in a future study is unlikely to be a concern in a future cluster randomised trial. We observed a greater proportion of participants who were working in the company for more than three years at baseline than we expected, and it is recommended that future questionnaires are designed with this in mind. For future studies in this population, we recommend that the ">3 years" category is subdivided into further categories to enable more precise data to be collected on the duration working in the company. For example, the questionnaire options could include "3–5 years", "5–10 years", and ">10 years" options, instead of just ">3 years".

The stepped wedge study design is unlikely to be a suitable design choice for a future cluster randomised trial conducted in call centres. We experienced difficulties in maintaining site interest (e.g. if some sites were randomised to receive the intervention 12 months later), and it was challenging to ensure that each site adhered to the scheduled data collection time points. Instead of a stepped wedge trial design, we suggest that a parallel-group cluster randomised trial could be used, where sites are recruited in pairs (or groups) over time.

## Conclusions

SUH is an adaptive, flexible, multi-component intervention that allows contact centres to develop a range of activities to suit their culture and context. The process evaluation findings

showed that the intervention was acceptable and feasible to deliver, and most contact centres implemented several actives aligning with every level of the programme's theory of change. Perceived subjective benefits such as reduced sedentary behaviour, increased physical activity, and improved staff morale and mood were reported by contact centre staff and managers. After the COVID-19 pandemic, most organisations have adopted a hybrid working format, and further research is required to understand and develop appropriate activities for this new work environment. The SUH programme shows potential in reducing sedentary behaviour and positively impacting several health and wellbeing outcomes in contact centres, and can now proceed to large-scale evaluation.

## Supporting information

**S1 Checklist. The consort checklist for pilot and feasibility studies.**
(PDF)

**S1 Protocol. Stand UP for health protocol.**
(DOCX)

**S1 File. Stand Up for Health programme details.**
(DOCX)

**S2 File. Post-lockdown individual activity plan sample.**
(PDF)

**S3 File. ActivPAL information and logbook to capture work and sleep times.**
(DOCX)

**S4 File. Progression criteria.**
(DOCX)

**S1 Fig. CONSORT flow diagram for post-lockdown data collection.**
(DOCX)

## Acknowledgments

We would like to sincerely thank everyone who contributed to this study:

Thanks to Ipsos MORI Scotland, especially Wayne Gilbert and Eilidh Gordon who made the pilot research project possible, and have been supporters of Stand Up for Health from the outset.

Masters Students who came up with the initial SUH concept- Audrey Buelo, Christina Katan, Laura Tirman, Florence Ashdown, Ruth Miller, and Isis Guerrero Castillo.

Contact centres who took part in the research. We cannot mention you by name due to maintain confidentiality, but we appreciate all the time and support you have given to the study.

Sit-Stand.com who provided desk risers and anti-fatigue mats to participating centres free of cost. Special thanks to Rik Mistry and Nicola Davenport from Sit-Stand.com for their help and enthusiasm.

Additional research assistance was provided by the following researchers and research students- Bradley MacDonald, Kieran Turner Lara Tarvit, Matthew Northcote, Zhen (Leo) Yang, Yueshun (Shaun) Shi, Daichun (Ellen) Zhan, Fariha Mosaddeque, Justyna Kedziera, Hannah Houghton, Ioannis Psomadakis-Karastamatis, Li Jiahang.

The Study Steering Committee members: Dr. Alison Kirk, University of Strathclyde, Dr. Claire Fitzsimons, University of Edinburgh, Ms. Laura Mandefield, University of York, Mr.

Wayne Gilbert, Associate Director, Ipsos MORI (Leith, Edinburgh), Ms. Eilidh Gordon, Administrator, Operations, Ipsos MORI (Leith, Edinburgh), Mr. Graeme Russell, CWU Regional BT safety chair, Ms. Sharon Currie, NHS Health Scotland.

## Author Contributions

**Conceptualization:** Divya Sivaramakrishnan, Graham Baker, Richard A. Parker, Jillian Manner, Scott Lloyd, Ruth Jepson.

**Data curation:** Divya Sivaramakrishnan, Jillian Manner, Ruth Jepson.

**Formal analysis:** Divya Sivaramakrishnan, Graham Baker, Richard A. Parker, Jillian Manner, Ruth Jepson.

**Funding acquisition:** Graham Baker, Richard A. Parker, Ruth Jepson.

**Investigation:** Divya Sivaramakrishnan, Jillian Manner, Ruth Jepson.

**Methodology:** Divya Sivaramakrishnan, Graham Baker, Richard A. Parker, Jillian Manner, Scott Lloyd, Ruth Jepson.

**Project administration:** Divya Sivaramakrishnan, Jillian Manner, Ruth Jepson.

**Writing – original draft:** Divya Sivaramakrishnan.

**Writing – review & editing:** Divya Sivaramakrishnan, Graham Baker, Richard A. Parker, Jillian Manner, Scott Lloyd, Ruth Jepson.

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
