## [Decision Letter · Decision Letter 0]

10 Jul 2023

PONE-D-23-13160A Mixed Method Evaluation of a Theory Based Intervention To Reduce Sedentary Behaviour In Contact Centres- the Stand Up For Health Stepped Wedge Feasibility StudyPLOS ONE

Dear Dr. Sivaramakrishnan,

Thank you for submitting your manuscript to PLOS ONE. After careful consideration, we feel that it has merit but does not fully meet PLOS ONE’s publication criteria as it currently stands. Therefore, we invite you to submit a revised version of the manuscript that addresses the points raised during the review process.

We look forward to receiving your revised manuscript.

Kind regards,

Alejandro Vega-Muñoz, Ph.D.

Academic Editor

PLOS ONE

3. We noted in your submission details that a portion of your manuscript may have been presented or published elsewhere. [A full evaluation report for the study has been published previously as required by NIHR- https://doi.org/10.3310/IEXP0277. We have acknowledged this in the paper and referenced any tables/figures adapted from the report. The manuscript has been developed and reviewed by the research team as a separate piece of writing suitable for publication in a journal, and includes several new figures, tables, and discussions. The presentation of results has also been modified to focus on specific research objectives. The paper has drawn out and presented a succinct summary of a rigorous outcome and process evaluation, and we believe that it would be relevant and useful to the research community.] Please clarify whether this publication was peer-reviewed and formally published. If this work was previously peer-reviewed and published, in the cover letter please provide the reason that this work does not constitute dual publication and should be included in the current manuscript.

Reviewers' comments:

Reviewer's Responses to Questions

**Comments to the Author**

1. Is the manuscript technically sound, and do the data support the conclusions?

Reviewer #1: Yes

Reviewer #2: Partly

2. Has the statistical analysis been performed appropriately and rigorously? 

Reviewer #1: N/A

Reviewer #2: No

3. Have the authors made all data underlying the findings in their manuscript fully available?

Reviewer #1: Yes

Reviewer #2: Yes

4. Is the manuscript presented in an intelligible fashion and written in standard English?

Reviewer #1: Yes

Reviewer #2: Yes

5. Review Comments to the Author

Reviewer #1: The topic addressed is very interesting and relevant to the current reality. Undoubtedly, the nature of work often predisposes workers to sedentary behaviors, so it is an issue that should be addressed, given the consequences that this can have on people.

In the methods, with regard to the participants, a better characterization of these should be provided. On the other hand, it is advisable to provide a solid justification, based on the available evidence, for the choice of the tools and/or instruments used.

The article should provide more bibliographical support to the discussion.

Reviewer #2: Summary

It is necessary to deepen the methodology studied with respect to statistical issues (Bivariate, etc.), how to validate that within the results the study is acceptable and feasible, as well as to deepen more relevant conclusions, and lines of future work.

Introduction

There are 10 citations for two lines, some sentences that are sentences end without reference example page 5 (Line n°1 and n°2), I do not see the research questions and neither the hypotheses. It is important to see if there are studies in the country, and the occupational sector where the study was carried out and to name it.

Theoretical framework.

Cases of bibliography of service companies are mentioned. There is a lack of case studies on what is being studied in the country.

Methodology

Regarding the control group to see the differences, I also lack papers that support this type of methodology, in addition to a greater type of analysis with bivariate techniques and/or multifactorial analysis.

Results.

The whole section is missing based on the methodology explained, there is not much coherence, there is a lack of order.

Bibliography.

Only 20 authors in the last 5 years of the 49 authors, it is very outdated, much more is missing.

6. PLOS authors have the option to publish the peer review history of their article (what does this mean?). If published, this will include your full peer review and any attached files.

Reviewer #1: No

Reviewer #2: No

---

## [Author Response · Author response to Decision Letter 0]

10 Sep 2023

We thank the reviewers for their comments. We have attached a document "Response to reviewers.docx" that includes a point-by-point response to all comments. We have attached a revised document with the changes and relevant sections highlighted (Revised Article with Changes Highlighted.docx) and have provided a clean copy as well (Revised Manuscript.docx).

Please do let us know if you have any additional recommendations for us. We would be pleased to revisit any of the comments and revise the manuscript accordingly.

We look forward to hearing from you regarding the revised manuscript.

Thank you and kind regards,

Divya Sivaramakrishnan (on behalf of all authors)

---

## [Decision Letter · Decision Letter 1]

17 Oct 2023

A Mixed Method Evaluation of a Theory Based Intervention To Reduce Sedentary Behaviour In Contact Centres- the Stand Up For Health Stepped Wedge Feasibility Study

PONE-D-23-13160R1

Dear Dr. Sivaramakrishnan,

We’re pleased to inform you that your manuscript has been judged scientifically suitable for publication and will be formally accepted for publication once it meets all outstanding technical requirements.

Kind regards,

Alejandro Vega-Muñoz, Ph.D.

Academic Editor

PLOS ONE

Reviewers' comments:

Reviewer's Responses to Questions

**Comments to the Author**

1. If the authors have adequately addressed your comments raised in a previous round of review and you feel that this manuscript is now acceptable for publication, you may indicate that here to bypass the “Comments to the Author” section, enter your conflict of interest statement in the “Confidential to Editor” section, and submit your "Accept" recommendation.

Reviewer #1: All comments have been addressed

Reviewer #2: All comments have been addressed

2. Is the manuscript technically sound, and do the data support the conclusions?

Reviewer #1: Yes

Reviewer #2: Yes

3. Has the statistical analysis been performed appropriately and rigorously? 

Reviewer #1: Yes

Reviewer #2: Yes

4. Have the authors made all data underlying the findings in their manuscript fully available?

Reviewer #1: Yes

Reviewer #2: Yes

5. Is the manuscript presented in an intelligible fashion and written in standard English?

Reviewer #1: Yes

Reviewer #2: Yes

6. Review Comments to the Author

Reviewer #1: I reiterate the relevance of the topic for the world of work. Thank you for responding and clarifying each of the points raised.

Reviewer #2: The revisions were worked on by the research team, both in the abstract, introduction, update of the Theoretical Framework, Methodology, Results, Conclusions and Bibliography with more contemporary authors. more contemporary authors.

Thank you very much.

7. PLOS authors have the option to publish the peer review history of their article (what does this mean?). If published, this will include your full peer review and any attached files.

Reviewer #1: No

Reviewer #2: No

---

## [Editor Report · Acceptance letter]

31 Oct 2023

PONE-D-23-13160R1 

A Mixed Method Evaluation of a Theory Based Intervention To Reduce Sedentary Behaviour In Contact Centres- the Stand Up For Health Stepped Wedge Feasibility Study 

Dear Dr. Sivaramakrishnan:

I'm pleased to inform you that your manuscript has been deemed suitable for publication in PLOS ONE. Congratulations! Your manuscript is now with our production department. 

Kind regards, 

on behalf of

Dr. Alejandro Vega-Muñoz 

Academic Editor

PLOS ONE